# Proprotein Convertase Subtilisin Kexin Type 9 (PCSK9) Deletion but Not Inhibition of Extracellular PCSK9 Reduces Infarct Sizes Ex Vivo but Not In Vivo

**DOI:** 10.3390/ijms23126512

**Published:** 2022-06-10

**Authors:** Rolf Schreckenberg, Annemarie Wolf, Tamara Szabados, Kamilla Gömöri, István Adorján Szabó, Gergely Ágoston, Gábor Brenner, Péter Bencsik, Péter Ferdinandy, Rainer Schulz, Klaus-Dieter Schlüter

**Affiliations:** 1Institute of Physiology, Faculty of Medicine, Justus-Liebig University, Gießen, 35390 Gießen, Germany; Rolf.Schreckenberg@physiologie.med.uni-giessen.de (R.S.); annema.wolf@t-online.de (A.W.); Rainer.Schulz@physiologie.med.uni-giessen.de (R.S.); 2Cardiovascular Research Group, Department of Pharmacology and Pharmacotherapy, Albert Szent-Györgyi Medical School, University of Szeged, 6720 Szeged, Hungary; szabados.tamara@med.u-szeged.hu (T.S.); kamilla.gomori@pharmahungary.com (K.G.); sz.istvan.adorjan@gmail.com (I.A.S.); agoston.gergely@med.u-szeged.hu (G.Á.); peter.bencsik@pharmahungary.com (P.B.); 3Pharmahungary Group, 6722 Szeged, Hungary; brenner.gabor@med.semmelweis-univ.hu (G.B.); peter.ferdinandy@pharmahungary.com (P.F.); 4Department of Pharmacology and Phamacotherapy, Faculty of Medicine, Semmelweis University, 1089 Budapest, Hungary

**Keywords:** cardioprotection, MMP9, ischemia/reperfusion

## Abstract

Hypoxia upregulates PCSK9 expression in the heart, and PCSK9 affects the function of myocytes. This study aimed to investigate the impact of PCSK9 on reperfusion injury in rats and mice fed normal or high-fat diets. Either the genetic knockout of PCSK9 (mice) or the antagonism of circulating PCSK9 via Pep2-8 (mice and rats) was used. Isolated perfused hearts were exposed to 45 min of ischemia followed by 120 min of reperfusion. In vivo, mice were fed normal or high-fat diets (2% cholesterol) for eight weeks prior to coronary artery occlusion (45 min of ischemia) and reperfusion (120 min). Ischemia/reperfusion upregulates PCSK9 expression (rats and mice) and releases it into the perfusate. The inhibition of extracellular PCSK9 does not affect infarct sizes or functional recovery. However, genetic deletion largely reduces infarct size and improves post-ischemic recovery in mice ex vivo but not in vivo. A high-fat diet reduced the survival rate during ischemia and reperfusion, but in a PCSK9-independent manner that was associated with increased plasma matrix metalloproteinase (MMP)9 activity. PCSK9 deletion, but not the inhibition of extracellular PCSK9, reduces infarct sizes in ex vivo hearts, but this effect is overridden in vivo by factors such as MMP9.

## 1. Introduction

Cardiovascular diseases (CVDs) are leading causes of death worldwide [1]. In addition to hypertension, diabetes, age, sex, alcohol and nicotine abuse, hypercholesterolemia is another risk factor for developing cardiovascular disease [2]. Therefore, an efficient reduction in hypercholesterolemia is crucial for improvements in CVD mortality and morbidity rates. The inhibition of proprotein kinase subtilisin/kexin type 9 (PCSK9) is the most efficient method to reduce low-density lipoprotein-cholesterol (LDL-C), a major transport form of cholesterol [3]. PCSK9 negatively regulates the LDL-receptor abundance in the membrane of hepatocytes and thus increases serum LDL-C [4], and PCSK9 efficacy is increased through binding to high-density lipoproteins [5].

Apart from its role in hepatic LDL clearance, the extrahepatic functions of PCSK9 have been described [6,7,8,9,10], and as such, an increase in platelet activity via PCSK9 might contribute to an increased incidence and progression of acute myocardial infarction (AMI) [11]. In the heart, PCSK9 is expressed in ventricular cardiomyocytes and is released by them. This contributes to the oxidized LDL (oxLDL)-induced impairment of cardiomyocyte cell shortening [12]. In line with this, the chronic treatment of mice with oxLDL increases the left ventricular diameter and the cardiomyocyte cross sectional area and reduces the ejection fraction, which are all effects that are antagonized by the administration of evolocumab, a PCSK9 antibody. The signaling of oxLDL involves LOX-1 (lectine-like oxidized low-density lipoprotein receptor-1) and dynamin-related protein 1 (Drp1), with the latter increasing mitochondria-induced oxidative stress [13]. Furthermore, PCSK9 deficiency may support heart failure [14].

PCSK9 may also play a role in ischemia/reperfusion (I/R) injury. This assumption is based on studies that have shown an upregulation of PCSK9 under hypoxic conditions in neonatal mouse cardiomyocytes [15]. Indeed, serum PCSK9 is elevated in rats and in patients suffering from AMI [16,17]. Two in vivo studies experimentally investigated the impact of PCSK9 on infarct remodeling in mice or infarct size in rats [15,18]. The first study suggested that PCSK9 negatively impacts post-infarct remodeling, whereas the second study showed that the pharmacological inhibition of PCSK9 prior to I/R reduces infarct size. The questions that could not be addressed in these in vivo studies are first, whether cardiac-derived PCSK9 participates in reperfusion injury, as in both cases, cardiac-derived PCSK9 interferes with hepatically released PCSK9, and second, whether under high-fat diet conditions, thereby under pathophysiological conditions normally associated with high PCSK9 levels, the inhibition of PCSK9 is still suitable to reduce infarct sizes. Under clinical conditions, the plasma levels of PCSK9 are inversely associated with the six-month left ventricular ejection fraction [19].

The impact of a high-fat diet on I/R injury is not clear, as it may decrease infarct size (“high-fat diet paradox”), increase infarct size or it may be neutral [3,20,21]. Hypercholesterolemia does not selectively lead to the upregulation of PCSK9. It favors inflammation and induces the expression of matrix metalloprotease (MMP)9 [22]. MMP9 is a commonly found biomarker associated with cardiac arrhythmias [22]. Moreover, it may directly support arrhythmia via the downregulation of connexin-43 [23]. Finally, angiotensin II can induce the expression of MMP9 [24] and PCSK9, and angiotensin II can act synergistically [25]. Therefore, we also addressed the question of whether PCSK9 interferes with MMP9 activity in I/R.

To address some of the open questions regarding PCSK9 and reperfusion injury, we investigated whether the genetic deletion of PCSK9 or the pharmaceutical inhibition of PCSK9 via Pep2-8 affects I/R injury in rodents ex vivo and in vivo. The ex vivo models allowed us to judge the direct role of PCSK9 in the heart, whereas the in vivo section of the study highlighted the more complex interaction between circulating PCSK9, which is mainly derived from the liver, local effects and interaction with MMP9. The latter aspect is of specific interest in the context of high-fat diets.

## 2. Results

### 2.1. Influence of I/R on PCSK9 mRNA Expression, Release and Contribution on I/R Injury in Rats Ex Vivo

First, we investigated the effect of ischemia duration on the reperfusion-induced expression of PCSK9 mRNA in rat hearts. The expression of PCSK9 mRNA was increased with the prolongation of ischemia in reperfused rat hearts (120 min of reperfusion each; Figure 1A). Given the greater ischemia-dependent damage in the 60 min group and the strong induction of PCSK9 after 45 min of ischemia, we used 45 min of flow arrest for the subsequent analysis of PCSK9 release into the perfusate and the modification of post-ischemic recovery via the administration of a PCSK9 inhibitor. The mRNA expression of PCSK9 induced a time-dependent release of PCSK9 from rat hearts into the perfusate (Figure 1B). Similarly, PCSK9 mRNA expression was increased in rat hearts undergoing ischemia/reperfusion in vivo in a transient way (Figure 1C). Collectively, this analysis shows that ischemic events increase the cardiac expression of PCSK9 and subsequently affect the local concentration of extracellular PCSK9. Whether this leads to the modification of post-ischemic recovery and infarct size was addressed in subsequent experiments.

To block a potential effect of the cardiac-dependent release of PCSK9 on cardiac function, Pep2-8 was added as a PCSK9 inhibitor to the perfusate prior to the onset of ischemia. As expected from our previous study [26], the addition of a PCSK9 inhibitor increased inotropy (as indicated by increased left ventricular developed pressure (LVDP) and rate pressure product (RPP), but not chronotropy (Figure 2A) in pre-ischemic hearts). This expected effect showed that the concentration of Pep2-8 we used here (10 µM) is functionally relevant in this setup. However, the inhibitor did not affect the functional recovery of the post-ischemic hearts (Figure 2B), the infarct size (Figure 2C) or the prevalence of edema as estimated by heart wet weight to body weight (Figure 2D). Although we confirmed an increased expression and secretion of PCSK9 during ischemia/reperfusion, we could not provide any evidence that locally released PCSK9 affects I/R injury. However, the study design only allowed conclusions regarding extracellular PCSK9 and did not rule out the possibility that intracellular PCSK9 may affect I/R injury. To address this point, we used the genetic deletion of PCSK9 by using PCSK9^−/−^ mice.

### 2.2. Effects of I/R on PCSK9 Expression and Contribution to I/R in Mice Ex Vivo

The I/R-dependent upregulation of PCSK9 expression was conformed in mice. As shown for rat hearts, 45 min of flow arrest was sufficient to induce the expression of PCSK9 in post-ischemic mouse hearts as well (Figure 3A). The amount of PCSK9 in the perfusate of the mouse hearts was, however, below the detection level of the assay. When we repeated the experiments with the PCSK9 inhibitor Pep2-8 with mouse hearts, as previously conducted with rat hearts, again, we found no effect on functional recovery (Figure 3B), infarct size (Figure 3C) or edema (Figure 3D) by the addition of the inhibitor. In contrast, the genetic deletion of PCSK9 improved functional recovery (Figure 3B), reduced infarct sizes (Figure 3C), but did not affect edema formation (Figure 3D). This part of the study again negates a role of extracellular PCSK9 in the context of I/R but suggests that by interfering with intracellular targets, PCSK9 contributes to reperfusion injury. However, the situation in vivo is more complex, as PCSK9 plasma concentration mainly reflects the secretion of PCSK9 from the liver and cardiac responsiveness affected by other plasma factors as well. Therefore, it was necessary to transfer the promising ex vivo mouse model to the in vivo situation to determine whether the genetic deletion of PCSK9 protects the heart in vivo as well.

### 2.3. Effects of I/R on PCSK9 Plasma Concentration and Contribution of PCSK9 to I/R Injury In Vivo under Normal Diet and High-Fat Diet

As PCSK9 antagonism was more efficient in patients with acute coronary syndrome when they displayed metabolic risk factors (32), we investigated the effect of the genetic deletion of PCSK9 on I/R injury in vivo in mice fed normal diets or high-fat diets. As previously described by others [27], the plasma levels of PCSK9 declined in mice fed high-cholesterol diets (Figure 4A). In contrast to the aforementioned ex vivo effect, the genetic deletion of PCSK9 did not affect infarct sizes in vivo (Figure 4B,C). To validate these results, we used two independent approaches to quantify infarct sizes: triphenyl tetrazolium (TTC) staining (Figure 3B) and serum TnI concentration (Figure 4C). No differences were observed between all groups with TTC staining, but the TnI assay suggested a slightly higher level of tissue damage in the high-fat diet group (Figure 4C). Two-way ANOVA with a Tukey test indicated diet was an independent variable determining TnI levels (*p* = 0.003).

However, the most obvious difference between the normal diet and high-fat diet was obtained for post-infarct survival. Whereas all mice undergoing 120 min of reperfusion survived until scarification, only 28.5% of mice fed a high-fat diet survived (Figure 5A). There was no effect of PCSK9 deletion on survival; however, during early reperfusion, there was a trend toward delayed death in mice with the deletion of PCSK9 (Figure 5B). The serum concentration of PCSK9 between survivors and non-survivors was not different (Figure 5C). The data suggest that PCSK9 does not affect I/R injury in vivo. This contrasts the protection seen ex vivo and indicates that other factors strongly affect reperfusion injury and outmatch the protective effect of PCSK9 inhibition specifically in the high-fat diet group. MMP9 is a suitable candidate that affects reperfusion injury among other factors. Indeed, the activity of MMP9 was strongly increased in the high-fat diet group irrespective of PCSK9 (Figure 5D).

## 3. Discussion

This study aimed to investigate the potential role of PCSK9 in I/R injury. Previous studies suggested that I/R injury is associated with plasma levels of PCSK9 and also that I/R increases the local expression of PCSK9 in the heart. The outcome of our study validated findings that reported the ischemia-dependent upregulation of PCSK9 but identified three major problems when considering the inhibition of PCSK9 as a tool to improve post-ischemic recovery. First, the inhibition of extracellular PCSK9 affected neither infarct size nor functional recovery in rats or mice ex vivo. Second, the genetic deletion of PCSK9 was cardioprotective, but this effect was lost in vivo. Third, although mice fed high-fat diets had lower levels of PCSK9, they had higher lethality during reperfusion, which is completely independent of PCSK9 but associated with MMP9 activity. Collectively, although we confirmed the induction of PCSK9 in ischemic hearts and cardiac protection by the deletion of PCSK9 ex vivo, we could not translate this effect to the in vivo setting. In the context of high-fat diets, we identified the upregulation of MMP9 as a confounder associated with increased mortality but no association with PCSK9.

In our previous study, we demonstrated that adult ventricular cardiomyocytes constitutively express and secrete PCSK9 [12,26]. Moreover, secreted PCSK9 co-activates signal transduction pathways that depress cardiac function [14,26]. In line with these ex vivo studies, an inverse relationship between PCSK9 and cardiac function, quantified as left ventricular ejection fraction (LVEF), was shown [26]. It had been reported that hypoxia and myocardial infarction upregulates PCSK9 in cardiac tissues [15,16,28]. In vivo, inflammation has been demonstrated in liver cells, which may also support such an upregulation of PCSK9 [29]. These findings suggest that the inhibition of circulating PCSK9 will improve post-ischemic recovery either by reducing infarct size or by the inhibition of signal pathways interfering with inotropy in survived cardiomyocytes or by a combination thereof. As previously conducted by others [18], we used Pep2-8 to inhibit extracellular PCSK9 and gave the inhibitor prior to flow arrest. The difference between both studies is that we used the inhibitor ex vivo, therefore targeting cardiac-derived PCSK9, whereas in vivo, the inhibitor also targeted hepatic-derived PCSK9. As predicted from our previous study [26], Pep2-8 increased left ventricular function. However, it affected neither infarct sizes nor functional recovery in rats or mice. This negative outcome was not predicted from the in vivo study in which Pep2-8 reduced infarct size and improved functional recovery in rats [18]. This negative outcome of the ex vivo experiment with Pep2-8 was contrasted by ex vivo experiments with hearts from mice with a genetic deletion of PCSK9. In this case, we found a strong reduction in infarct size and also an improvement in functional recovery. Similarly, by the permanent ligature of the heart, it was shown that the genetic deletion of PCSK9 reduces necrosis and improves function [15]. However, such an experiment tested the impact of PCSK9 on post-infarct remodeling rather than I/R injury.

Whereas the ex vivo part of our study confirmed basic findings from previous studies, such as the induction of the cardiac expression of PCSK9 under ischemia and cardiac protection by the deletion of PCSK9 as in post-infarct remodeling in vivo, we could not translate these promising results to the more complex in vivo situation. This is the first study that used PCSK9^−/−^ mice to study I/R injury. An important point needs to be addressed. The observed discrepancy between ex vivo and in vivo experiments draws our attention to the differences between ex vivo and in vivo blood. Most importantly, in vivo blood transports unnumbered hormones and cytokines that affect infarct size and cardiac function, whereas ex vivo blood is replaced by a saline medium lacking such factors. Therefore, we conclude that the protective effect seen by a lack of endogenously expressed PCSK9 on I/R injury is superposed by non-cardiac factors affecting I/R injury. Irrespective of these findings excluding PCSK9 as a player in acute I/R injury, the data do not rule out the possibility that PCSK9 affects early remodeling in cardiac tissue based on autophagy, as mentioned earlier [30].

The combination of genetic PCSK9 deletion and a high-fat diet allowed us to investigate such interactions in more detail. In agreement with other studies [22], we found increased MMP9 activity in mice fed a high-fat diet. MMP9 has been associated with cardiac arrhythmia [22]. During I/R, the heart is exposed to a high risk of arrhythmia. The combination of high MMP9 activity and I/R turned out to be lethal for most of the mice. This main effect of a high-fat diet was not affected by PCSK9.

In summary, our study negates PCSK9 as a promising target to treat reperfusion injury despite its cardioprotective potential in specific settings. However, our study is in line with clinical data showing that low PCSK9 activity is associated with low LDL-C and a lower risk of myocardial infarction [31,32,33,34]. Nevertheless, this may be an LDL-dependent effect rather than a direct PCSK9 effect.

## 4. Materials and Methods

### 4.1. Animals

Adult Wistar rats (female, 200–225 g) were ordered from Janvier Labs (Le Genest Saint Isles, France). PCSK9 knockout mice (B6;129S6-Pcsk9^tm1Jdh^*,* 25–30 g) and appropriate controls (B6129SF2/J, 8–9 weeks old) were supplied from The Jackson Laboratory (Maine, USA). Mice were housed according to the Guide for the Care and Use of Laboratory Animals (NIH Publication No. 85–23, revised 1996). All protocols were approved by the Justus-Liebig-University Giessen (permission number: 666_M and 561_M), as well as the National Scientific Ethical Committee on Animal Experimentation Budapest, Hungary. In this study, only male mice were used to avoid the effect of sex on experiments. Mice and rats were randomized to the different groups. This study included experiments with rat and mouse hearts. These were carried out to show that the observed effects of PCSK9 are species-independent.

### 4.2. I/R Ex Vivo Procedures

Mice: Mice were anaesthetized with isoflurane inhalation (5%) and killed via cervical dislocation. Hearts were quickly prepared and attached to an Aortic Cannula (Ø 1 mm, Hugo Sachs Elektronik-Harvard Apparatus, March, Germany). The cannulated heart was then connected to a Langendorff perfusion system (Hugo Sachs Elektronik-Harvard Apparatus, March, Germany), where it was perfused retrograde with a 37 °C warm modified Krebs–Henseleit buffer (NaCl 118 mM, KCl 4.7 mM, MgSO_4_ 0.8 mM, KH_2_PO_4_ 1.2 mM, G/glucose 5 mM, CaCl_2_ 2.5 mM, NaHCO_3_ 25 mM and pyruvate 1.9 mM; pH 7.4; sterile-filtered with an 0.2 µm filter; saturated with 95% oxygen and 5% carbogen). The perfusion pressure was adjusted to 70 mmHg (transduced by a Replacement Transducer Head for APT300 Pressure Transducer, Hugo Sachs Elektronik-Harvard Apparatus) and kept constant during the whole experiment. A small balloon (built up from cling-film), which was connected to a pressure transducer (Combitrans 1-fach Set Mod.II University Giessen, B. Braun, Melsungen, Germany), was carefully inserted into the left ventricle and inflated up to 12–14 mmHg (end diastolic pressure) to assess left ventricular function. Additionally, the hearts were paced at 600 bpm to ensure a stable heart rate. Left ventricular pressure (LVPD = systolic pressure-diastolic pressure) was determined. After a stabilization phase of 5 min, the perfusion and pacing were switched off for 45 min to generate a no-flow ischemia. Thereafter, the heart was reperfused for another 120 min. Normoxic control hearts were perfused for 170 min.

To validate a possible function of PCSK9 in the progress of I/R, the PCSK9 inhibitor Pep2-8 was solved in Langendorff buffer. This solution was washed for 5 min before ischemia and during the first 10 min of reperfusion. Pep2-8, which is the smallest available chemical Pcsk9 inhibitor, was supplied from Tocris Bioscience (Bristol, UK) and used at a concentration of 10 µM according to ref. [26,35].

At the end of the experiment, the heart was taken off the Langendorff perfusion system and stored at −80 °C or further processed for TTC staining (described below).

Rats: The left ventricular function of isolated rat hearts was evaluated as described before [36]. Hearts were excised as mentioned above and attached to a 16-gauge needle. The cannulated heart was then connected to a Langendorff perfusion system, where it was perfused retrograde with an oxygenated saline buffer (NaCl 140 mM, NaHCO_3_ 24 mM, KCl 2.7 mM, NaH_2_PO_4_ 0.4 mM, MgCl_2_ 1 mM, CaCl_2_ 1.8 mM and glucose 5.0 mM, pH 7.4). A polyvinylchloride balloon, which was connected to a pressure transducer, was carefully inserted into the left ventricle to assess left ventricular pressure (LVDP). At the end of the experiment, the heart was taken off the Langendorff perfusion system and stored at −80 °C. Initial experiments were performed with 30, 45 and 60 min of flow arrest and subsequent reperfusion for 2 h. Thereafter, all further experiments were performed with 45 min of flow arrest.

Additionally, Langendorff perfusates were collected at two different time points (1st and 10th minute of reperfusion) and stored at −20 °C for subsequent analysis with a commercially available PCSK9 ELISA kit purchased by Cusabio Technology LLC, Wuhan, China (Rat PCSK9 ELISA kit) and used following the instructions. The intra- and inter-assay variability of the ELISA was calculated to a coefficient of variability of 12.8% and 4.5%, respectively.

### 4.3. I/R Experimental Protocols

PCR analysis of rat hearts (experiments shown in Figure 1) was performed with samples from previous studies [37,38]. Here, the analysis was extended to the mRNA expression of PCSK9 and to the analysis of the PCSK9 concentration in the perfusate. The latter was important to control whether mRNA induction was translated into PCSK9 release and gave the basis for newly performed rat heart experiments described in Figure 2.

The experiments described in Figure 2 were newly performed according to the established protocol (see experiments in Figure 1 and I/R ex vivo procedures). The n number of rat hearts was calculated based on the assumption that LVDP recovery differs more than 25% and with an expected standard deviation taken from similar experiments in recent years [39].

The experiments described in Figure 3 were performed on mouse hearts and were newly performed for this study. Here, we used *n* = 6 mice per group based on the assumption that infarct size will be reduced by 33% and with an expected standard deviation taken from similar experiments in recent years [40].

### 4.4. I/R In Vivo

PCSK9^−/−^ and B6129SF2/J (PCSK9^+/+^) mice were fed with a high-fat diet (2% cholesterol and 0.5% cholic acid, Table 1) for 8 weeks prior to I/R experiments.

I/R was induced in vivo using LAD occlusion, as described before [41]. Briefly, mice were anesthetized with sodium pentobarbital and intubated for ventilation. The chest was opened to visualize the left coronary artery at the 4th intercostal space. A string (8-0 Prolene) was positioned around the left anterior descending branch (LAD) of the left anterior coronary artery. To ensure LAD occlusion (45 min) as well as subsequent release (120 min of reperfusion), the string was looped, and a small piece of a plastic cannula was placed onto the coronary artery before the loop was pulled tight.

Experiments were initially planned with *n* = 10 mice per group. However, due to the high mortality rate during reperfusion in the high-fat diet group, we increased the group size to 21 mice to receive at least 6 mice per group that reached the planned end of the reperfusion (120 min) to quantify infarct sizes. Serum samples were taken and analyzed as described before [42].

### 4.5. TTC Staining

To determine the infarct size after I/R, ventricles were frozen for 30 min at −20 °C directly after the experiment. Subsequently, the frozen heart was cut into 8–10 slices. which were then incubated in 1.2% triphenyl-tetrazolium chloride (TTC) for 30 min at room temperature. Heart slices were stored in PBS overnight, and digital images were taken with a M60 microscope (2.5-fold magnification, Leica, Wetzlar, Germany). Images were analyzed with Leica Application Suite LAS (Leica, version 4.12) to planimetrically determine infarct sizes.

### 4.6. RNA Isolation and qRT-PCR

RNA was isolated from left ventricular tissue by using peqGold Trifast (peqlab, Biotechnologie GmbH, Germany) according to the manufacturer’s protocol. To avoid genomic DNA contamination, samples were treated with DNase (1 U/µg RNA; Invitrogen, Karslruhe, Germany) for 15 min at 37 °C. The reverse transcription of RNA (1 µg/10 µL) into cDNA was performed with Superscript RNaseH reverse transcriptase (200 U/µg RNA; Invitrogen) and oligo dTs at 37 °C for 60 min. Quantitative Real-Time PCR was performed using the CFX Connect Real-Time PCR detection system (Bio-Rad, Munich, Germany) in combination with IQ SYBR green real-time supermix (Bio-Rad, Munich, Germany). Primers for PCSK9, HPRT, GAPDH and B2M (Table 2) were used, and quantification was performed by applying the ΔΔ Ct method [43].

### 4.7. MMP9 Activity Measurement

To measure MMP9 activity, gelatin zymography was performed from serum samples as described previously [36]. Briefly, 8% polyacrylamide gels were copolymerized with gelatin (2 mg/mL, type A from porcine skin; Sigma-Aldrich, Budapest, Hungary), and 50 µg of protein per lane was loaded. After electrophoresis, gels were washed and stained with 0.05% coomassie brilliant blue and subsequently destained in aqueous methanol–acetic acid.

### 4.8. Statistics

Data are expressed as box plots representing the Q25, median, Q75 and full range. The statistical comparison of two groups was performed with a two-sided-*t*-test if the normal distribution of samples could be verified using Levene’s test. Otherwise, the Mann–Whitney-U test was applied. The comparison of more than two groups was performed with one-way ANOVA following a post hoc analysis with the Student–Newman–Keuls test. *p*-levels < 0.05 were regarded as significant and indicated as an asterisk or as expressed in the figure’s legend.

## 5. Conclusions

In conclusion, our study negates PCSK9 as a promising target to treat reperfusion injury despite its cardioprotective potential in specific settings. However, our study is in line with clinical data showing that low PCSK9 activity is associated with low LDL-C and a lower risk of myocardial infarction [31,32]. Nevertheless, this may be an LDL-dependent effect rather than a direct PCSK9 effect.

## Figures and Tables

**Figure 1 ijms-23-06512-f001:**
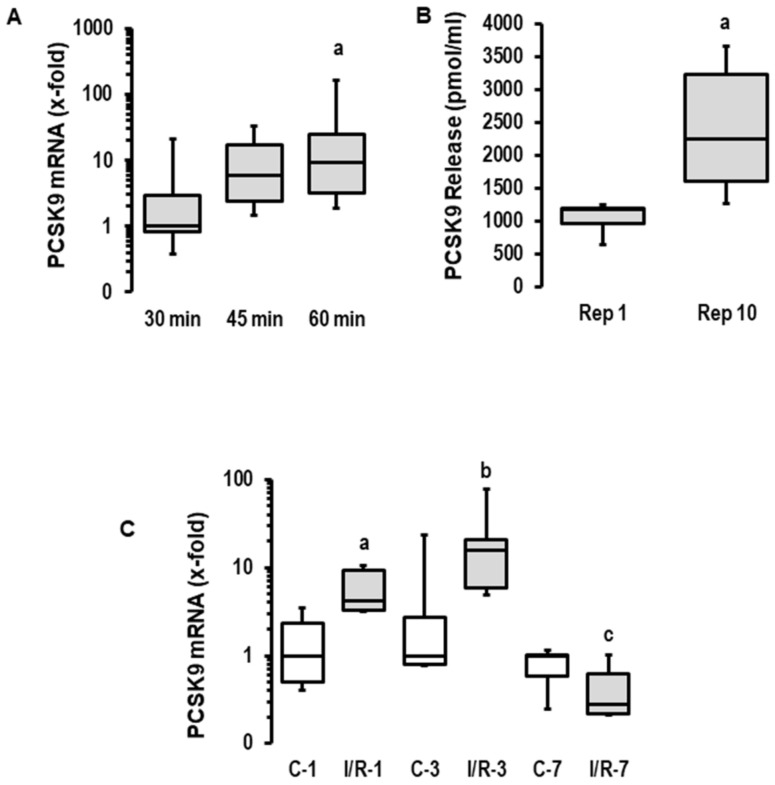
*Expression, secretion and effect of PCSK9 on reperfusion injury in rat hearts.* (**A**) Rat hearts were exposed to 30, 45 and 60 min of ischemia and 120 min of reperfusion (I/R). PCSK9 mRNA expressions of left ventricles were analyzed (*n* = 8 each; *p* = 0.026 for Kruskal–Wallis Test; a; *p* = 0.010 vs. 30 min (Bonferroni test with correction for multiple testing). (**B**) Rat hearts were exposed to 45 min of ischemia, and PCSK9 protein was quantified in the perfusate (*n* = 8 each; a, *p* = 0.009 vs. Rep. 1; Student’s *t*-test for unpaired samples). (**C**) PCSK9 mRNA expressions of the left ventricles of sham operated rats (**C**) or rats exposed to 45 min of ischemia and reperfusion (I/R). Recovery was analyzed after 1, 3 and 7 days (*n* = 5 each; a; *p* = 0.028 I/R vs. (**C**); b; *p* = 0.076 I/R vs. (**C**); c; *p* = 0.209 I/R vs. (**C**); data are full ranges (whiskers) with median and 25 and 75% quartiles (boxes).

**Figure 2 ijms-23-06512-f002:**
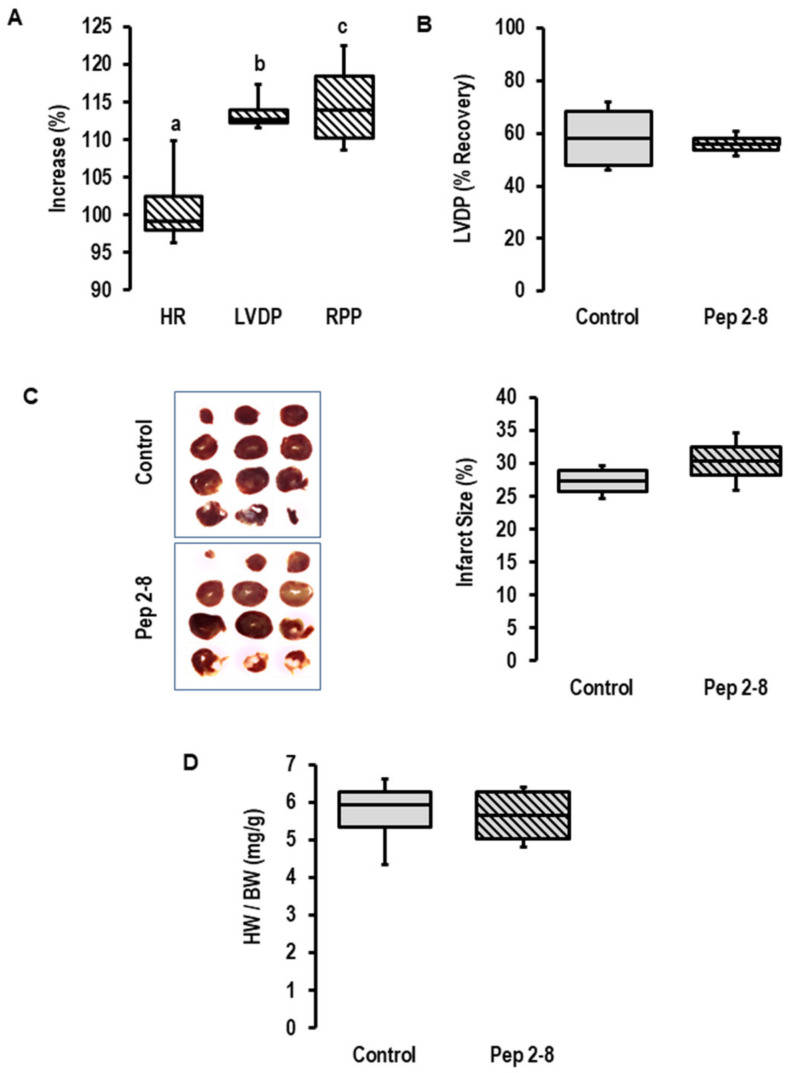
*Effect of PCSK9 inhibition on post-ischemic recovery*. (**A**) Effect of Pep2-8 (10 µM, striated bars) on heart rate (HR), left ventricular developed pressure (LVDP) and rate pressure product (RPP) in the pre-ischemic period (*n* = 4 each; a; *p* = 0.738 vs. pre-treatment; b; *p* = 0.002 vs. pre-treatment; c; *p* = 0.018 vs. pre-treatment; Student’s *t*-test for paired samples). (**B**) Effect of Pep2-8 on left ventricular developed pressure (LVDP) after 120 min of reperfusion, expressed as % of pre-ischemic values; *p* = 0.728 vs. control; Student’s *t*-test for unpaired samples. (**C**) Effect of Pep2-8 on infarct size (left original registration, right quantification (*p* = 0.215 vs. control; Student’s *t*-test for unpaired samples). (**D**) Effect of Pep2-8 on heart weight to body weight ratio (*p* = 0.923 vs. control; Student’s *t*-test for unpaired samples). Data are full ranges (whiskers) with median and 25 and 75% quartiles (boxes).

**Figure 3 ijms-23-06512-f003:**
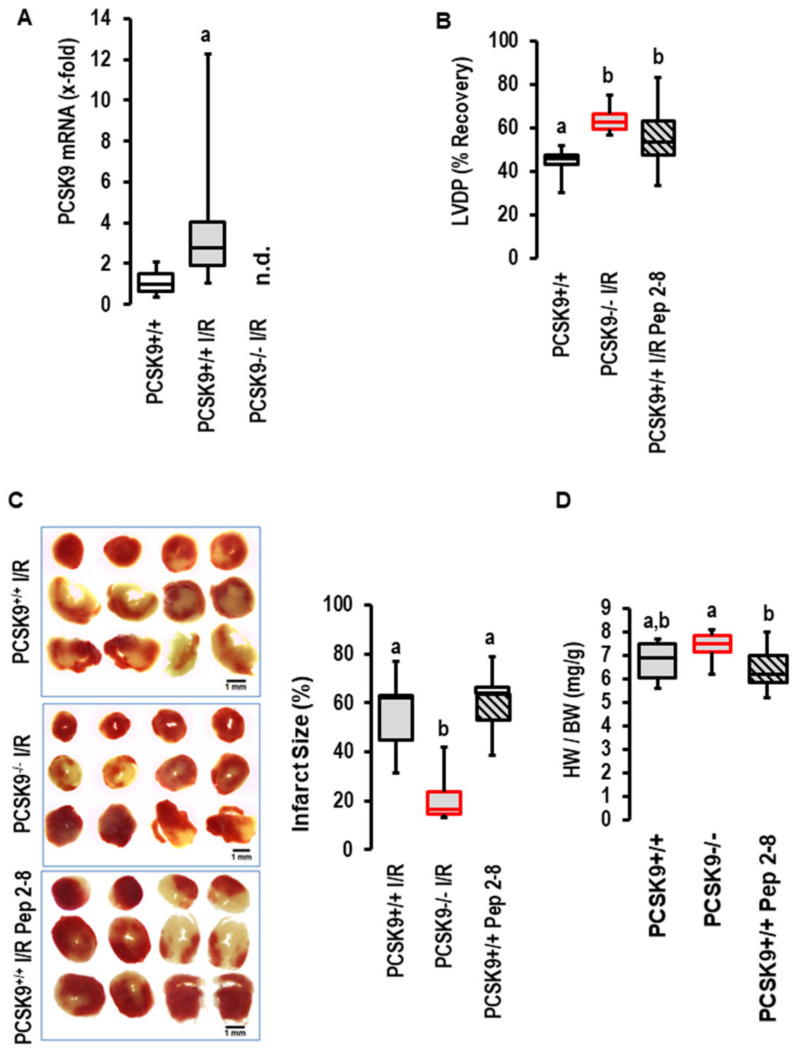
*Expression and effect of PCSK9 inhibition on reperfusion injury in mouse hearts.* (**A**) Expression of PCSK9 mRNA in the left ventricles of normoxic controls (*n* = 10) and mouse hearts exposed to ischemia/reperfusion (*n* = 8); a, *p* = 0.008 vs. control; Student’s *t*-test for unpaired samples. (**B**) Effect of Pep2-8 and genetic deletion of PCSK9 on left ventricular developed pressure (LVDP) expressed as % recovery of pre-ischemic values. One-Way ANOVA *p* = 0.001; Student–Newman–Keuls indicated groups between knockout and Pep2-8 vs. wild-type controls as indicated by letters. (**C**) Effect of Pep2-8 and genetic deletion of PCSK9 on infarct sizes with original registration on left and quantification on right; 1-Way ANOVA *p* = 0.003; Student–Newman–Keuls indicated groups between knockout and Pep2-8 vs. wild-type controls as indicated by letters (a,b). (*n* = 5 each). (**D**) Effect of Pep2-8 and genetic deletion of PCSK9 on heart weight to body weight ratio; 1-Way ANOVA *p* = 0.043; Student–Newman–Keuls indicated groups between knockout and Pep2-8 vs. knockouts as indicated by letters.; *n* = 10 each. Data are full ranges (whiskers) with median and 25 and 75% quartiles (boxes).

**Figure 4 ijms-23-06512-f004:**
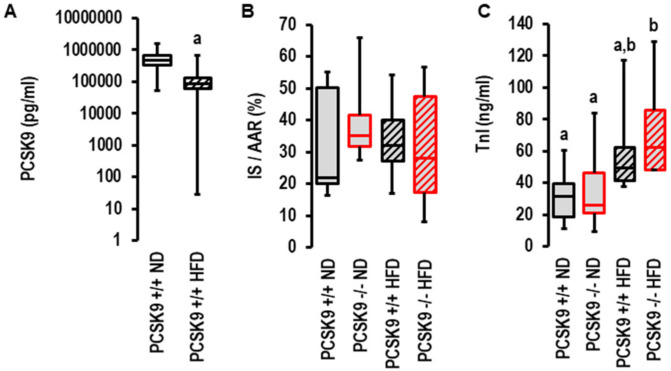
*Effect of PCSK9 genetic deletion and high-fat diet on serum PCSK9 levels and infarct sizes* in vivo. (**A**) Effect of high-fat diet (HFD, *n* = 30) on serum PCSK9 levels compared to normal diet (ND, *n* = 8); a; *p* = 0.003; Student’s *t*-test for unpaired samples. Non different groups are indicated by identical letters (a,b). (**B**) Inarct sizes (IS) normalized to area of risk (AAR) analyzed using 1-Way ANOVA (*p* = 0.949) and 2-Way ANOVA (*p* = 0.951 for genotype; *p* = 0.892 for diet; *p* = 0.587 for interaction; HFD: *n* = 10 each; ND: *n* = 6 each. (**C**) Infarct sizes measured as troponin I release in serum samples analyzed using 1-Way ANOVA (*p* = 0.022) and 2-Way ANOVA with *p* = 0.330 for genotype, *p* = 0.003 for diet, and *p* = 0.646 for interaction; ND: *n* = 8 (wild-type) and *n* = 9 (knockout); HFD: *n* = 6 (wild type) and *n* = 5 (knockout). Data are full ranges (whiskers) with median and 25 and 75% quartiles (boxes).

**Figure 5 ijms-23-06512-f005:**
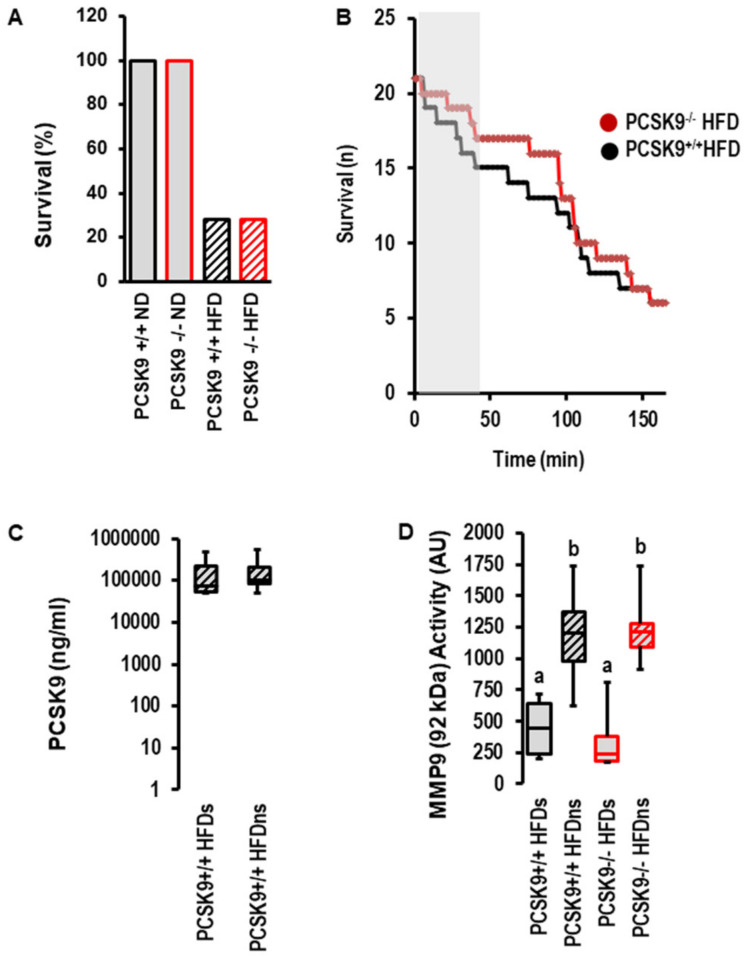
*Effect of PCSK9 and high-fat diet (HFD, n = 21) on survival, PCSK9 serum levels and MMP9 activity.* (**A**) Percent survival of the different groups. (**B**) Survival curve for high-fat diet mice during ischemia and reperfusion. Kaplan–Meier analysis *p* = 0.497 (Log-Rank Test). (**C**) Plasma levels in HFD mice and comparison between survivors (s; *n* = 6) and non-survivors (ns; *n* = 14; *p* = 0.953). (**D**) MMP9 activity in serum samples from mice as indicated; 1-Way ANOVA: a vs. b; *p* = 0.0001; 2-Way-ANOAV: *p* = 0.0001 for survival; *p* = 0.726 for genotype; *p* = 0.404 for interaction; survivors (s; *n* = 6 each); non-survivors (ns; *n* = 14 PCSK9^+/+^ and n = 15 PCSK9^−/−^). Data are full ranges (whiskers) with median and 25 and 75% quartiles (boxes).

**Table 1 ijms-23-06512-t001:** Composition of the mouse chow.

Dry matter	%	88
Crude protein	%	22.04
Crude fat	%	3.35
Crude fiber	%	6.34
Crude ash	%	9.31
Lyisine	%	0.99
Methionine	%	0.4
Methionine + cystine	%	0.78
Calcium	%	0.81
Phosphorous	%	0.66
Sodium	%	0.17
Vitamin A	IU/kg	12,250
Vitamin D3	IU/kg	1800
Vitamin E	mg/kg	61

The chow was complemented with 2% cholesterol and 0.5% cholic acid.

**Table 2 ijms-23-06512-t002:** List of primers used in this study.

Gene	Forward	Reverse
*B2M*	GCTATCCAGAAAACCCCTCAA	CATGTCTCGATCCCAGTAGACGGT
*GAPDH*	ACGGCACAGTCAAGGCCGAG	CACCCTTCAAGTGGGCCCCG
*HPRT*	CCA GCG TCG TGA TTA GCG AT	CAA GTC TTT CAG TCC TGT CC
*PCSK9*	CACCATGGGCACCGTCAGCTCCAG	AAACTGGAGCTCCTGGGAGGCC

## Data Availability

The data that support the findings of this study are available from the corresponding author upon reasonable request.

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
