# Peer review of "Proprotein Convertase Subtilisin Kexin Type 9 (PCSK9) Deletion but Not Inhibition of Extracellular PCSK9 Reduces Infarct Sizes Ex Vivo but Not In Vivo"

_ijms, 2022, doi:10.3390/ijms23126512_

Round 1

Reviewer 1 Report

This is an interesting paper which evalueated the role of PCSK9 in I/R injury using mouse and rat Langendorff perfused heart. The authors came to a conclusion that only genetic PCSK9 deletion but not extracellular inhibition of this convertase can effectively induced infarct size.  As I understand they explanain it by different actions of secreted vs intracellular PCSK9. Both are obviously gone upon genetic deletion, whereas the used inhibitor only blokcs extracellular convertase.

1. The authors used 10 µM of Pep2-8 for extracellular inhibition. However, I could not find clear and convincing experimental or even accurately discussed historical evidence that this inhibitor selectively affects extracellular PCSK9 at the used concentration with sufficient specificity. It is so clear? Can also unspecific effects be involved? If only extracellular proteins are affected how comes the effect of cardiac contractility which the authors used to conclude that inhibitor "worked". I would really like to see a "waterproof" clear demonstation or at least conclusive discussion on previous work that PCSK9 was biochemically inhibited extra- but not intracellularly. 

2. Please check the statistics used in individual figures. The methods section clearly states multuple comparison test which should be used after one-way ANOVA when comparing more then two groups or when having more than one comparison in the same diagramm. Figure legends themselves mostly refer to T-test where ANOVA with postdoc test for multiple comparison should have been used.

3. There are still quite many typos and instances of "awkward language" use in the ms. I recommend to spell check carefully and improve the writing style where possible.

Author Response

We thank the reviewer for her/his comments and suggestions. Please read our point-by-point responses.

You wrote: “1. The authors used 10 µM of Pep2-8 for extracellular inhibition. However, I could not find clear and convincing experimental or even accurately discussed historical evidence that this inhibitor selectively affects extracellular PCSK9 at the used concentration with sufficient specificity. It is so clear? Can also unspecific effects be involved? If only extracellular proteins are affected how comes the effect of cardiac contractility which the authors used to conclude that inhibitor "worked". I would really like to see a "waterproof" clear demonstration or at least conclusive discussion on previous work that PCSK9 was biochemically inhibited extra- but not intracellularly.”

Response:  We apologize if the reason and rational to use the inhibitor has not been described before in a very clear way. Of course, the use of the inhibitor was done on the basis of previous studies (doi 10.1007/s00395-020-00814-w; open access). Our basic experiments are conducted on cultured isolated adult rat ventricular cardiomyocytes which were exposed to recombinant PCSK9. The source of recombinant PCSK9 was HepG2 cells overexpressing PCSK9 (with LacZ controls). The supernatants of the culture media were then exposed to cultured adult rat ventricular cardiomyocytes and load-free cell shortening was measured. In these studies, Pep 2-8 (10 µmol/L) and antibodies directed against PCSK9 (alirocumab) were used to antagonize the effect of PCSK9 on cell function. In contrast, if PCSK9 was expressed directly in cardiomyocytes (with LacZ controls) the effect on cell shortening was neutral, despite a strong increase in cellular expression. Therefore, we showed previously very clear that extracellular PCSK9 affects the function of cardiomyocytes but not intracellular PCSK9 and that Pep 2-8 (10 µmol/L) was sufficient to block the effect of extracellular PCSK9 but not that of intracellular. On the molecular level it is known that the inhibitor blocks the catalytic domain of PCSK9 (doi 10.1074/jbc.M113.514067, open access).

You wrote: “2. Please check the statistics used in individual figures. The methods section clearly states multuple comparison test which should be used after one-way ANOVA when comparing more then two groups or when having more than one comparison in the same diagramm. Figure legends themselves mostly refer to T-test where ANOVA with postdoc test for multiple comparison should have been used.”

Response: We thank the reviewer for the advice and have carefully checked the description of statistics.

Figure 1: In 1A, the data for 30 min and 60 min a not normally distributed (Shapiro-Wilk Test). Therefore, we used a Kruskall-Wallis-Test for multiple comparison with Bonferroni post-hoc analysis (with correction fort multiple testing). The legend has been changed properly. In Fig. 1B and Fig 1C two groups are compared as indicated.

Figure 2A-D: In all cases we compared Pep 2-8 treatment with controls (2 groups).

Figure 3: 3A: As PCSK9 was not detectable in knockouts, we have a two-group comparison as indicated.  3B: We have corrected statistical analysis: 1-Way-ANOVA p=0.001; Student-Newman-Keuls indicated groups between knockout and Pep 2-8 vs. wild-type controls. Figure corrected. 3C: We have corrected statistical analysis: 1-Way-ANOVA p=0.003; Student-Newman-Keuls indicated groups between knockout and Pep 2-8 vs. wild-type controls. Figure corrected. 3D: We have corrected statistical analysis: 1-Way-ANOVA p=0.043; Student-Newman-Keuls indicated groups between knockout and Pep 2-8 vs. knockouts. Figure corrected.

Figure 4 and 5: No corrections in statistical analysis required.

You wrote: “3. There are still quite many typos and instances of "awkward language" use in the ms. I recommend to spell check carefully and improve the writing style where possible.”

Response: We thank the reviewer for the advice and have carefully checked the manuscript.

Reviewer 2 Report

The manuscript is of interest. Relative to the effects of PCSK9 beyond low-density lipoprotein receptor regulation, I suggest to quote more recent reviews instead of reference 6 dated 2016. Please consider Am J Pathol. 2021 Aug;191(8):1385-1397 and Endocr Rev. 2022 May 12;43(3):558-582. Conversely, the Authors cannot ignore the manuscript by Da Dalt on PCSK9 and heart failure (Eur Heart J. 2021 Aug 21;42(32):3078-3090). Again, the Authors missed to mention that inflammation rise the expression of PCSK9 (J Biol Chem. 2016 Feb 12;291(7):3508-19).

The impact of Pep2-8 on the circulating levels of PCSK9 is mandatory. Is the inhibition given by Pep2-8 similar to that obtained by using PCSK9-/- mice. It is not the same. Selective inhibition of PCSK9 in the heart impacts mitochondrial activity (discussed according to the literature). More info has to be reported on the ELISA kit used to evaluate the circulating levels of PCSK9 in rats (not common). Please give intra and inter assays.

Figures are of low quality and not easily readable for this reviewer. Please change style and choose one uniform way to present the data, e.g., coloured bars. Moreover, figure legends have to be consistent throughout that manuscript, namely, if you report the statistical analysis in panel B, it should be valid for the other panels.

Discussion has to be improved, namely, which is the impact of PCSK9 inhibiton in ACS?

Author Response

We thank the reviewer for her/his comments and suggestions. Please read our point-by-point responses.

You wrote: “The manuscript is of interest. Relative to the effects of PCSK9 beyond low-density lipoprotein receptor regulation, I suggest to quote more recent reviews instead of reference 6 dated 2016. Please consider Am J Pathol. 2021 Aug;191(8):1385-1397 and Endocr Rev. 2022 May 12;43(3):558-582. Conversely, the Authors cannot ignore the manuscript by Da Dalt on PCSK9 and heart failure (Eur Heart J. 2021 Aug 21;42(32):3078-3090). Again, the Authors missed to mention that inflammation rise the expression of PCSK9 (J Biol Chem. 2016 Feb 12;291(7):3508-19).”

Response: We thank for the comment and have included these references into the manuscript. However, we did not ignore the manuscript by Da Dalt. However, as in I/R an induction of PCSK9 is discussed (see Introduction), a PCSK9 deficiency is linked to HFpEF. Neither does HFpEF be related to I/R injury, nor does a deficiency mimic the induction. Therefore, we did not cite this manuscript. This does not mean that the study itself is not important. Here, we completely agree to the reviewers opinion as there are up-coming data indicating that a certain level of PCSK9 is properly required. In any way, the study now cites the reference. Similarly, the important study on the regulation of PCSK9 expression in hepatic cells by pro-inflammation is an important reference. However, our study deals with extra-hepatic cells, inflammation in the ex vivo models is largely ruled out (replacement of circulating blood by saline perfusate), and finally acute effects (2 h reperfusion) rather than chronic effects are analyzed. In any way, the study now cites the reference. See refs. 9, 10, 14, 29.  

You wrote: “The impact of Pep2-8 on the circulating levels of PCSK9 is mandatory. Is the inhibition given by Pep2-8 similar to that obtained by using PCSK9-/- mice. It is not the same. Selective inhibition of PCSK9 in the heart impacts mitochondrial activity (discussed according to the literature). More info has to be reported on the ELISA kit used to evaluate the circulating levels of PCSK9 in rats (not common). Please give intra and inter assays.”

Response: We thank for the comment. The use of the inhibitor was done on the basis of previous studies (doi 10.1007/s00395-020-00814-w; open access). Our basic experiments are conducted on cultured isolated adult rat ventricular cardiomyocytes which were exposed to recombinant PCSK9. The source of recombinant PCSK9 was HepG2 cells overexpressing PCSK9 (with LacZ controls). The supernatants of the culture media were then exposed to cultured adult rat ventricular cardiomyocytes and load-free cell shortening was measured. In these studies, Pep 2-8 (10 µmol/L) and antibodies directed against PCSK9 (alirocumab) were used to antagonize the effect of PCSK9 on cell function. In contrast, if PCSK9 was expressed directly in cardiomyocytes (with LacZ controls) the effect on cell shortening was neutral, despite a strong increase in cellular expression. Therefore, we showed previously very clear that extracellular PCSK9 affects the function of cardiomyocytes but not intracellular PCSK9 and that Pep 2-8 (10 µmol/L) was sufficient to block the effect of extracellular PCSK9 but not that of intracellular. On the molecular level it is known that the inhibitor blocks the catalytic domain of PCSK9 (doi 10.1074/jbc.M113.514067, open access). In our experiments it was important that Pep 2-8 increased pre-ischemic function of ex vivo hearts, indicating that the concentration used here is functionally active (Fig. 2A).

We extended our discussion suggesting that impairment of mitochondrial function by selective inhibition of PCSK9 may impact mitochondria.

Finally, we add data to intra- and inter assay variability for the PCSK9 ELISA (Coefficients of Variabilty: 12.8% and 4.5%, respectively).

You wrote: “Figures are of low quality and not easily readable for this reviewer. Please change style and choose one uniform way to present the data, e.g., coloured bars. Moreover, figure legends have to be consistent throughout that manuscript, namely, if you report the statistical analysis in panel B, it should be valid for the other panels.”

Response: We thank for your comment. For all figures we used identical symbols for identical treated groups. All sham controls are shown in white (Fig. 1C, 3A, 4A), hearts exposed to I/R are shown in gray (Fig. 1A; 1B, 1C; 2B, 2C, 2D, 3B, 3C, 3D, 4B, 4C, 5A, 5D), hearts treated with Pep 2-8 are shown with bars from the upper left to the right down (Fig. 3B, 3C, 3D), knockout mice are always shown in red boxes (Fig. 3B, 3C, 3D, 4B, 4C, 5A, 5D), high fat diet is always shown with bars from the left down to the upper right (Fig. 4B, 4C, 5A, 5C, 5D). Furthermore, all groups are labeled on the figure. We apologize for mistakes in Fig 3C and 5A. These have been corrected. The quality of the figures was lost by transfer into the word document. As we have to upload the original figures as well, we believed that during production process they are replaced by the originals. Nevertheless, we tried other options for incorporation of the figures in the word matrix and hope that you can now see the figures in a better way.

You wrote: “Discussion has to be improved, namely, which is the impact of PCSK9 inhibiton in ACS?”

Response:  We have improved the discussion and included additional studies dealing with PCSK9 and ACS (Refs. 30, 33, 34). However, please keep in mind that the high-fat diet used here is NOT a diabetes model whereas under clinical conditions diabetes is a major determinant improving outcome by inhibition of PCSK9, that the ex vivo experiments used here are in saline perfused hearts with no effect on inflammatory cells reducing the importance of inflammation, and that we analyzed the acute reperfusion injury, whereas clinical studies focus on long-term effects. Therefore we discussed our results in the original version strictly on the line of our experiments.

Round 2

Reviewer 2 Report

No further comments, although I believe tha reference 6 can be deleted